# Factors correlated with visual outcomes at two and four years after vitreous surgery for proliferative diabetic retinopathy

**Katsuhiro Nishi**[ID][☯], **Koichi Nishitsuka**[ID]*[☯], **Teiko Yamamoto**[¤], **Hidetoshi Yamashita**

Department of Ophthalmology and Visual Sciences, Yamagata University Faculty of Medicine, Yamagata, Yamagata, Japan

☯ These authors contributed equally to this work.
¤ Current address: Kanamecho Yamamoto Eye Clinic, Toshima-ku, Tokyo, Japan
* mlc12186@nifty.com

**Data Availability Statement:** All relevant data are within the paper and its Supporting Information files.

## Abstract

Proliferative diabetic retinopathy (PDR) is the most severe case of diabetic retinopathy that can cause visual impairment. This study aimed to reveal the factors correlated with better postoperative visual acuity after a long follow-up in patients who underwent vitrectomy for PDR. We retrospectively analyzed the data set including systemic findings, ocular findings, and surgical factors from registered patients who could be completely followed up for 2 or 4 years after vitrectomy. We ultimately enrolled 128 eyes from 100 patients who underwent vitrectomy for PDR between January 2008 and September 2012 and were followed up for >2 years. Among them, 91 eyes from 70 patients could be followed up for 4 years. Factors related to the postoperative visual acuity of $\geq$20/40 and $\geq$20/30 after 2 and 4 years were investigated by logistic regression analysis. Better postoperative visual acuity correlated with the following factors: no rubeosis iridis ([$\geq$20/40 at 2 years; odds ratio {OR}, 0.068; 95% confidence interval {CI}, 0.012–0.39; P = 0.003], [$\geq$20/30 at 2 years; OR, 0.07; 95% CI, 0.01–0.40; P = 0.03], [$\geq$20/30 at 4 years; OR, 0.078; 95% CI, 0.006–0.96; P = 0.04]), no fibrovascular membrane [($\geq$20/40 at 2 years; OR, 0.22; 95% CI, 0.061–0.81; P = 0.02), ($\geq$20/40 at 4 years; OR, 0.26; 95% CI, 0.07–0.94; P = 0.04), ($\geq$20/30 at 4 years; OR, 0.14; 95% CI, 0.04–0.52; P = 0.004)], existing vitreous hemorrhage ($\geq$20/30 at 2 years; OR, 9.55; 95% CI, 1.03–95.27; P = 0.04), and no reoperation ([$\geq$20/40 at 4 years; OR, 0.15; 95% CI, 0.03–0.78; P = 0.02], [$\geq$20/30 at 4 years; OR, 0.06; 95% CI, 0.07–0.54; P = 0.01]). Treatment provision before disease severity and treatment without complications were associated with good postoperative visual acuity.

## Introduction

Diabetic retinopathy is the leading cause of visual impairment worldwide [1]. In Japan, it accounts for 12.8% of all newly certified visual impairment cases [2]. Its most severe condition is proliferative diabetic retinopathy (PDR), which can cause visual acuity decrement or visual loss in patients with diabetic retinopathy [3,4]. Vitrectomy is indicated for nonclearing

**Funding:** This work was supported by JSPS
KAKENHI Grant Numbers JP25462704,
JP20K18373.

**Competing interests:** The authors have declared
that no competing interests exist.

vitreous hemorrhage (VH) or traction retinal detachment [5,6]. The clinical endpoints of PDR treatment were vitreoretinal lesion removal and blindness prevention.

Generally, patients with PDR undergo surgery in their 40s to 60s [7]. Considering that the average life expectancy of patients with diabetes mellitus (DM) is approximately 70 years, those who have undergone vitreous surgery for PDR will live roughly 10–30 years [8]. Hence, the goal for PDR treatment should not only to prevent blindness but also to maintain visual acuity after vitrectomy.

Although the factors correlated with the postoperative visual outcome of PDR have been reported extensively [7,9–11], the postoperative follow-up period varies in each case. To examine PDR cases more accurately, we need to continually assess the postoperative course as long as possible. Treatments for PDR should aim to improve as well as maintain patient's visual acuity. In this study, we aimed to build a data set that mainly included cases that could be completely followed up 2 or 4 years after a primary surgery and to reveal the factors correlated with the visual outcomes after vitrectomy for PDR.

## Materials and methods

This retrospective study was performed in accordance with the Declaration of Helsinki and approved by the Ethics Committee of the Yamagata University Faculty of Medicine (approval number: H26-21). All data were fully anonymized before we accessed them and the IRB waived the requirement for informed consent. This study investigated 147 eyes from 116 patients with PDR who underwent primary vitreous surgery at Yamagata University Hospital between January 2008 and September 2012. We retrospectively reviewed the medical records of these patients. A total of 106 eyes from 82 patients who had been attending only to Yamagata University Hospital were examined. We also collected the information of patients who had been attending to other hospitals after vitreous surgery. All patients with persistent VH and traction retinal detachment underwent three-port 20-gauge (G) pars plana vitrectomy or microincision vitreous surgery (MIVS) (23-G or 25-G); the 20-, 23-, and 25-G system was used for 71, 45, and 12 eyes, respectively (MIVS for 57 eyes). Two vitreoretinal surgeons performed all the surgical procedures. However, surgical cases of only diabetic macular edema were excluded. All patients did not receive anti-vascular endothelial growth factor (VEGF) therapy as a preoperative adjunct. Pan retinal photocoagulation was cautiously performed before or during vitrectomy on all patients. Participants were treated for vision-affecting lesions such as posterior capsular opacification, progressed cataract, neovascular glaucoma, and diabetic macular edema (DME) during the postoperative course.

### Data collected

The systemic factors collected were as follows: age, sex, duration from visual loss awareness to the primary vitreous surgery, hypertension history, DM duration, preoperative glycosylated hemoglobin (HbA1c), oral DM medication, insulin treatment, diabetic nephropathy history, coronary heart disease and/or stroke history, anticoagulant and/or antiplatelet agent administration, preoperative systolic and diastolic blood pressure, heart rate, and blood biochemical examination.

Moreover, the ophthalmologic findings were categorized into three sections: preoperative, intraoperative, and postoperative. The preoperative ophthalmologic findings were as follows: intraocular lens implantation, retinal photocoagulation, the history of intravitreal injection of triamcinolone acetonide, rubeosis iridis, ocular hypertension (>21 mmHg), VH, posterior vitreous detachment, fibrovascular membrane, retinal detachment, and macular detachment. The intraoperative ophthalmologic findings were the following: cataract surgery,

intraoperative retinal photocoagulation, gas tamponade, silicone oil tamponade, intraoperative complications (iatrogenic retinal break and retinal dialysis), and the number of used gauge (20-G or MIVS). Lastly, the postoperative ophthalmologic findings were as follows: reoperation and postoperative complications (VH, retinal detachment, and neovascular glaucoma).

For visual acuity measurement, we used a Japanese decimal visual acuity chart placed 5 m away from the patient. We measured decimal visual acuity preoperatively and 3 months, 6 months, 1 year, 2 years, 3 years, and 4 years after the primary vitreous surgery. The decimal visual acuity was converted into Snellen visual acuity and logarithmic minimum angle of resolution (logMAR) to examine visual acuity change.

We compared the decimal visual acuity distribution as well as the amount of vision change between the preoperative period and 2 years postoperatively and between 2 and 4 years postoperatively. An increase of ≥0.3 logMAR unit, a change of <0.3 logMAR unit, and a decrease of ≥0.3 logMAR unit in comparison with the preoperative value were defined as "improvement," "invariant," and "worsening," respectively. We then examined statistically the factors correlated with visual acuity of ≥20/40 Snellen (0.5 Japanese decimal visual acuity) or ≥20/30 Snellen (0.7 Japanese decimal visual acuity) at 2 and 4 years after a primary surgery for PDR. In Japan, driver's license can only be renewed if at least one eye has a visual acuity of ≥20/30.

## Statistical analysis

Fisher's exact test, chi-square test, Mann-Whitney $U$ test and analysis of variance (ANOVA) were used for the statistical analysis. For the factors of $P < 0.1$, we employed stepwise forward logistic regression analysis. For all the analyses, $P < 0.05$ was considered to be statically significant. All statistical data were analyzed using the PASW Statistics 18 (SPSS Inc., Chicago, IL, USA).

## Results

A total of 128 eyes from 100 patients (87.1%) could be followed up for >2 years after a primary surgery. In total, 91 eyes from 70 patients could be followed-up 4 years after the primary surgery. However, 37 eyes from 30 patients could not be evaluated 4 years after the primary surgery. Therefore, 37 eyes were categorized under the 2–3-year follow-up group. Tables 1 and 2 summarize the demographics of patients. Furthermore, 91 eyes were from 68 men, and 37 eyes were from 32 women. Their mean age was 55.7 ± 9.1 years. Regarding patients' age during the primary surgery, 29% (37 eyes) of the patients aged <50 years, whereas 61% (78 eyes) aged <60 years. No significant difference was found between the backgrounds of patients that could be followed up for 4 years and of those that could be followed up for 2–3 years postoperatively. In total, 11 patients presented with diabetic macular edema that required additional treatment after vitrectomy. During the research period, VEGF treatment was not approved in Japan, and patients were treated with retinal photocoagulation and local steroid. Seven eyes presented with neovascular glaucoma that required additional treatment after vitrectomy.

Comparison of the visual outcomes between preoperative and 2 and 4 years after primary vitreous surgery is illustrated in Fig 1. Both distributions of visual acuity at 2 and 4 years after primary vitreous surgery were more significantly improved than the distribution of preoperative visual acuity (P < 0.0001). The distribution of visual acuity between 2 and 4 years postoperatively was not significantly different (P = 0.59). Two years after surgery, the visual acuity values in 64.0% (82 of 128 eyes) and 39.8% (50 eyes of 128 eyes) of patients were ≥20/40 and ≥20/30, respectively. Four years after surgery, the visual acuity values in 53.8% (49 of 91 eyes) and 46.1% (42 of 91 eyes) of patients were ≥20/40 and ≥20/30, respectively.

**Table 1. Patient demographics.**

| | Total | 4 years follow-up | 2–3 years follow-up | P value[a] |
|---|---|---|---|---|
| | N = 128 | N = 91 | N = 37 | |
| Age (years) | 55.7 ± 9.1 | 55.1 ± 9.0 | 57.4 ± 9.1 | 0.29 |
| Sex (male) | 91 | 66 | 25 | 0.67 |
| Duration from visual loss awareness to the primary surgery (months) | 3.9 ± 3.8 | 4.3 ± 4.3 | 2.9 ± 2.6 | 0.62 |
| Hypertension | 76 | 53 | 23 | 0.84 |
| Diabetes mellitus duration (years) | 12.1 ± 6.8 | 12.1 ± 6.6 | 11.8 ± 7.3 | 0.7 |
| HbA1c (%) | 7.4 ± 1.3 | 7.3 ± 1.1 | 7.7 ± 1.5 | 0.35 |
| Oral medication for diabetes mellitus | 64 | 43 | 21 | 0.44 |
| Insulin treatment | 76 | 55 | 21 | 0.84 |
| Diabetic nephropathy | 88 | 57 | 31 | 0.12 |
| History of coronary heart disease and/or stroke | 23 | 16 | 7 | 1 |
| Anticoagulant and/or antiplatelet agent administration | 28 | 20 | 8 | 1 |
| Intraocular lens implantation | 38 | 25 | 13 | 0.4 |
| Preoperative retinal photocoagulation | 111 | 81 | 30 | 0.26 |
| Intravitreal injection of triamcinolone acetonide | 1 | 1 | 0 | 1 |
| Rubeosis iridis | 16 | 13 | 3 | 0.4 |
| Ocular hypertension | 10 | 8 | 2 | 0.72 |
| Vitreous hemorrhage | 102 | 73 | 29 | 1 |
| Posterior vitreous detachment | 32 | 22 | 10 | 0.48 |
| Fibrovascular membrane | 72 | 55 | 17 | 0.17 |
| Retinal detachment | 30 | 24 | 6 | 0.26 |
| Macular detachment | 18 | 14 | 4 | 0.78 |
| Cataract surgery | 63 | 45 | 18 | 1 |
| Intraoperative retinal photocoagulation | 111 | 76 | 35 | 0.15 |
| Gas tamponade | 24 | 20 | 4 | 0.21 |
| Silicone oil tamponade | 3 | 2 | 1 | 1 |
| Intraoperative complications | 15 | 10 | 5 | 1 |
| MIVS | 57 | 54 | 13 | 0.24 |
| Reoperation | 23 | 16 | 7 | 1 |
| Postoperative complications | 32 | 21 | 11 | 0.5 |

HbA1c: Glycosylated hemoglobin; BCVA: Best corrected visual acuity; logMAR: Logarithmic minimum angle of resolution; MIVS: Microincision vitrectomy surgery.

[a] Comparison of patients followed up for >4 years vs. 2–3 years.

The transition and change in the amount of postoperative visual acuity are depicted in Fig 2. As observed, the mean logMAR visual acuity changed during the preoperative period and 3 months, 6 months, 1 year, 2 years, 3 years, and 4 years postoperatively. Compared with the preoperative visual acuity, the visual activity in each subsequent observation point was significantly improved ($P < 0.0001$; ANOVA). Table 3 shows the distribution of visual acuity improvement, invariance, and worsening at 2 and 4 years postoperatively. At 2 years after surgery, the percentage of visual acuity improvement (logMAR $\leq$ −0.3), invariance (−0.3 < logMAR < 0.3), and worsening (logMAR $\geq$ 0.3) was 74.2% (95 of 128 eyes), 13.3% (17 of 128 eyes), and 12.5% (16 of 128 eyes), and at 4 years postoperatively, 69.2% (63 eyes of 91 eyes), 16.5% (15 eyes of 91 eyes), and 14.3% (13 eyes of 91 eyes) were obtained, respectively. Nonetheless, the distribution of visual acuity improvement, invariance, and worsening between 2 and 4 years after surgery was not significantly different ($P = 0.710$).

**Table 2. Demographics of systemic factors in patients.**

| | Total | 4 years follow-up | 2–3 years follow-up | P value[a] |
|---|---|---|---|---|
| | N = 128 | N = 91 | N = 37 | |
| Systolic BP (mmHg) | 138.9 ± 17.7 | 138.1 ± 17.7 | 140.9 ± 17.7 | 0.79 |
| Diastolic BP (mmHg) | 78.5 ± 11.5 | 78.0 ± 12.1 | 79.7 ± 10.0 | 0.63 |
| Heart rate (beats/min) | 74.8 ± 9.8 | 74.0 ± 10.2 | 77.0 ± 7.9 | 0.67 |
| TP (g/dl) | 6.9 ± 0.5 | 7.0 ± 0.5 | 6.9 ± 0.5 | 0.87 |
| Alb (g/dl) | 4.0 ± 0.4 | 4.0 ± 0.4 | 3.9 ± 0.5 | 0.63 |
| T.Bil (mg/dl) | 0.6 ± 0.2 | 0.6 ± 0.2 | 0.6 ± 0.1 | 0.26 |
| AST (IU/L) | 20.0 ± 6.3 | 20.4 ± 6.5 | 19.0 ± 5.7 | 0.42 |
| ALT (IU/L) | 23.6 ± 11.5 | 24.4 ± 12.3 | 21.6 ± 9.3 | 0.52 |
| LDH (IU/L) | 204.6 ± 34.8 | 205.3 ± 36.3 | 202.8 ± 31.3 | 0.95 |
| BUN (mg/dl) | 21.7 ± 8.7 | 21.5 ± 8.9 | 22.2 ± 8.4 | 0.63 |
| Crea (mg/dl) | 1.5 ± 1.0 | 1.5 ± 1.0 | 1.5 ± 0.9 | 0.73 |
| eGFR (ml/min/1.73 m$^2$) | 62.6 ± 27.0 | 64.1 ± 28.0 | 59.0 ± 24.5 | 0.54 |
| UA (mg/dl) | 5.8 ± 1.3 | 5.8 ± 1.3 | 5.8 ± 1.3 | 0.72 |
| Na (mEq/L) | 140.5 ± 1.8 | 140.6 ± 1.8 | 140.4 ± 1.9 | 0.55 |
| K (mEq/L) | 4.4 ± 0.4 | 4.4 ± 0.3 | 4.5 ± 0.4 | 0.46 |
| Cl (mEq/L) | 104.3 ± 2.5 | 104.2 ± 2.2 | 104.5 ± 3.3 | 0.96 |
| TG (mg/dl) | 175.6 ± 88.4 | 158.2 ± 70.7 | 221.0 ± 126.1 | 0.62 |
| TC (mg/dl) | 205.4 ± 39.1 | 201.6 ± 36.4 | 215.6 ± 45.4 | 0.25 |
| WBC (/μl) | 6708 ± 1564 | 6725 ± 1745 | 6668 ± 1128 | 0.6 |
| RBC (/μl) | 428 ± 53.1×10$^4$ | 430 ± 49.7 × 10$^4$ | 421 ± 61.4 × 10$^4$ | 0.62 |
| Hb (g/dl) | 12.7 ± 1.6 | 12.8 ± 1.5 | 12.6 ± 1.8 | 0.6 |
| Ht (%) | 38.2 ± 4.4 | 38.5 ± 3.9 | 37.5 ± 5.3 | 0.39 |
| Plat (/μl) | 23.0 ± 4.8 | 22.9 ± 5.1 | 23.4 ± 4.2 | 0.55 |
| PT (%) | 11.1 ± 0.6 | 11.0 ± 0.5 | 11.4 ± 0.9 | 0.28 |
| PT-INR | 0.93 ± 0.06 | 0.92 ± 0.05 | 0.98 ± 0.11 | 0.34 |
| APTT (seconds) | 28.9 ± 2.9 | 29.1 ± 2.9 | 28.2 ± 3.0 | 0.21 |

BP, blood pressure; TP, total protein; Alb, albumin; T.Bil, total bilirubin; AST, aspartate aminotransferase; ALT, alanine aminotransferase; LDH, lactate dehydrogenase; BUN, blood urea nitrogen; Crea, creatinine; eGFR, estimated glomerular filtration rate; UA, uric acid; Na, sodium; K, potassium; Cl, chlorine; TG, triglyceride; TC, total cholesterol; WBC, white blood cell; RBC, red blood cell; Hb, hemoglobin; Ht, hematocrit; Plat, platelet; PT, prothrombin time; PT-INR, prothrombin time–internationalized normalized ratio; APTT, activated partial thromboplastin time.

[a] Comparison of patients followed up for 4 years vs. 2–3 years.

Table 4 lists the factors correlated with the visual acuity of ≥20/40 at 2 years after surgery for PDR. Multivariate analysis showed that patients who achieved ≥20/40 at 2 years after the primary surgery for PDR had no rubeosis iridis (P = 0.003) and no fibrovascular membrane (P = 0.02) preoperatively.

Table 5 summarizes the factors correlated with the visual acuity of ≥20/30 at 2 years post-operatively. According to the multivariate analysis, patients who achieved ≥20/30 at 2 years after primary surgery for PDR had no rubeosis iridis (P = 0.03) but had VH (P = 0.04) preoperatively.

Table 6 presents the factors correlated with the visual acuity of ≥20/40 at 4 years postoperatively. Multivariate analysis showed that patients who achieved ≥20/40 at 4 years after surgery for PDR had no fibrovascular membrane (P = 0.04) and no reoperation (P = 0.02) preoperatively.

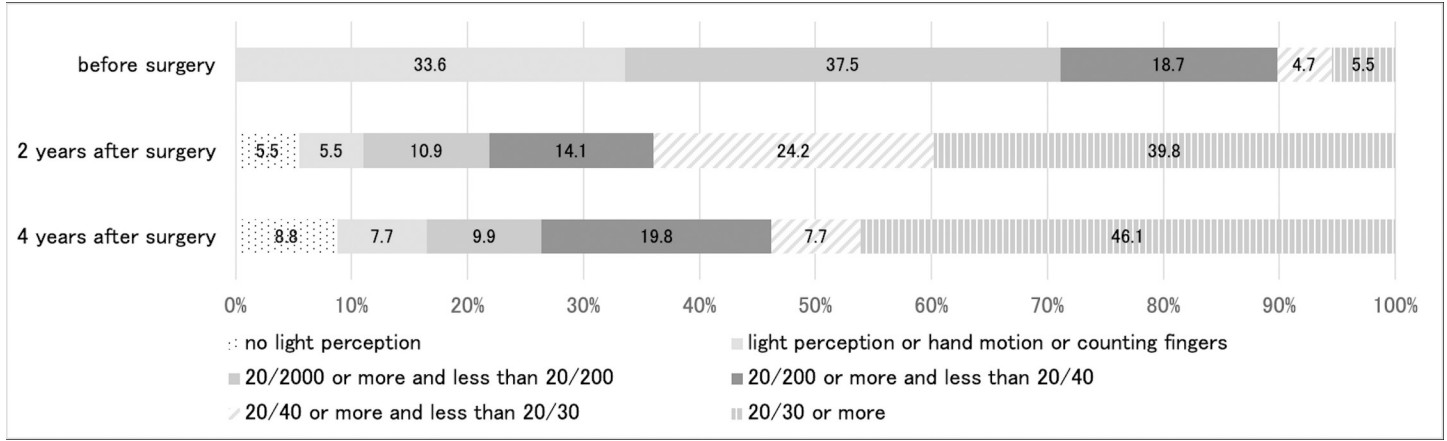

**Fig 1. Comparison of the visual outcomes between preoperative period and at 2 and 4 years after primary vitreous surgery.**

Table 7 presents the factors correlated with visual acuity of ≥20/30 at 4 years postoperatively. According to the multivariate analysis, patients who achieved ≥20/30 at 4 years after surgery for PDR had no rubeosis iridis (P = 0.04), no fibrovascular membrane (P = 0.004), and no reoperation (P = 0.01) preoperatively.

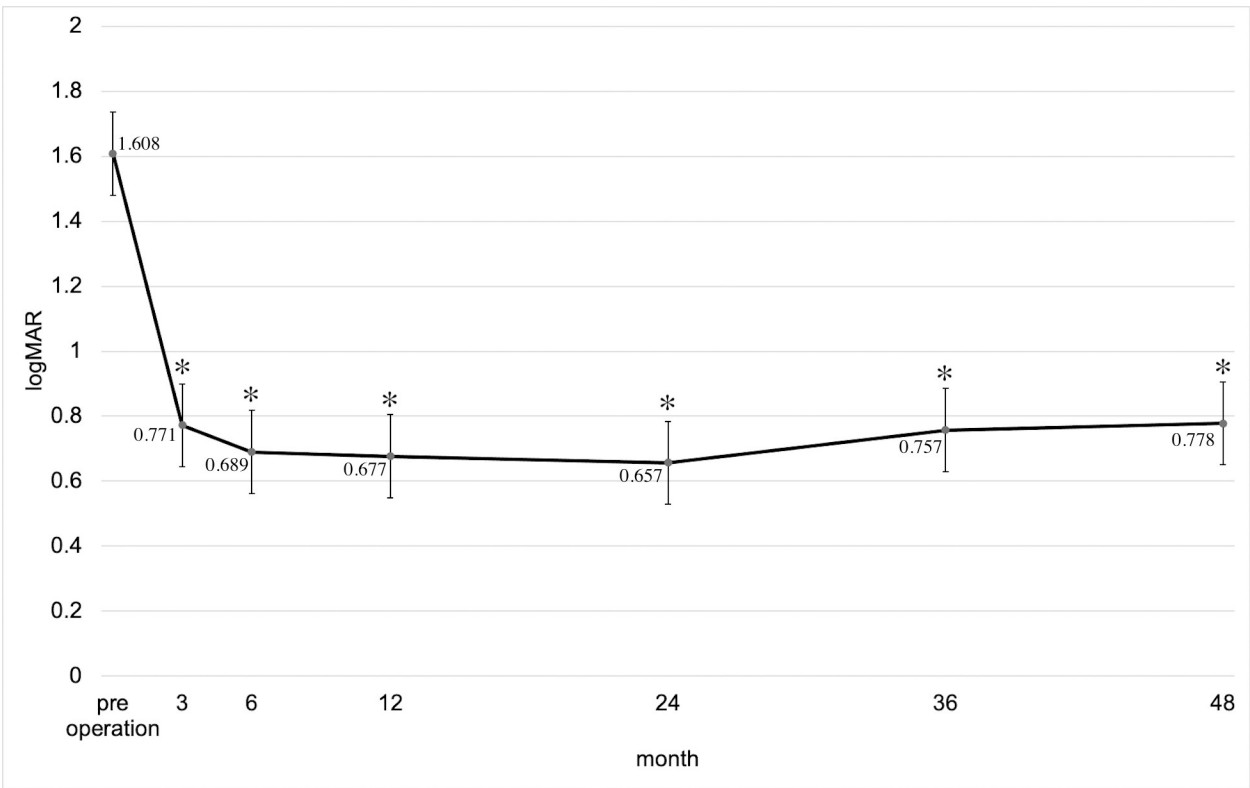

**Fig 2. Transition and change in the amount of postoperative visual acuity.** Mean BCVA (logMAR) ± standard error of the means (± SEM) during the preoperative period and 3 months, 6 months, 1 year, 2 years, 3 years, and 4 years after the primary vitrectomy. *P < 0.001 compared with preoperative mean BCVA (logMAR) based on the ANOVA test with Bonferroni correction.

**Table 3. Distribution of visual acuity improvement, invariance, and deterioration at 2 and 4 years after surgery.**

|                               | Improvement* | Invariant** | Worsen*** | P value |
|-------------------------------|-------------|-------------|-----------|---------|
| 2 years after surgery (N = 128) | 95 (74.2%) | 17(13.3%)  | 16(12.5%) | 0.710   |
| 4 years after surgery (N = 91)  | 63 (69.2%) | 15(16.5%)  | 13(14.3%) |         |

* Postoperative BCVA improved by ≥0.3 LogMAR unit compared with preoperative BCVA.

** Postoperative BCVA changed within 0.3 LogMAR unit compared with preoperative BCVA.

*** Postoperative BCVA worsened by ≥0.3 LogMAR unit compared with preoperative BCVA.

BCVA: Best corrected visual acuity; LogMAR: Logarithmic minimum angle of resolution.

## Discussion

This study demonstrated that the absence of rubeosis iridis and fibrovascular membrane before the surgery for PDR correlated with the visual acuity of ≥20/40 at 2 years postoperatively, whereas the absence of rubeosis iridis and the presence of VH preoperatively correlated with the visual acuity of ≥20/30 at 2 years postoperatively. At 4 years postoperatively, the preoperative absence of fibrovascular membrane and reoperation correlated with the visual acuity of ≥20/40, whereas the preoperative absence of rubeosis iridis, fibrovascular membrane, and reoperation correlated with ≥20/30. Furthermore, the distribution of visual acuity at 2 years after the primary vitreous surgery was more significantly improved than that of preoperative visual acuity. However, such distribution between 2 and 4 years postoperatively had no significant difference.

Rubeosis iridis is the neovascularization of the iris as a result of severe retinal ischemia. When neovascularization progresses to the angle between the iris, the intraocular pressure increases, leading to angiogenic glaucoma [12]. Many people who are in this condition have poor prognosis even after undergoing glaucoma surgery, often resulting in blindness [13]. Mason et al. [14] reported that preoperative and postoperative iris neovascularization, postoperative macular ischemia, and postoperative VH were risk factors for light perception or no light perception vision after vitrectomy in patients with diabetic retinopathy. In our study, the absence of preoperative rubeosis iridis is associated with visual acuities of ≥20/40 and ≥20/30 at 2 years and ≥20/30 at 4 years after the primary vitrectomy for PDR.

**Table 4. Analysis of the factors correlated with the visual acuity of ≥20/40 at 2 years after surgery.**

|                                        | Univariate analysis | Multivariate analysis |            |         |
|----------------------------------------|---------------------|-----------------------|------------|---------|
|                                        | P value             | OR                    | 95% CI     | P value |
| Rubeosis iridis                        | 0.001               | 0.068                 | 0.012–0.39 | 0.003   |
| Ocular hypertension                    | 0.004               | 0.42                  | 0.048–3.71 | 0.43    |
| Vitreous hemorrhage                    | 0.001               | 0.86                  | 0.24–3.01  | 0.81    |
| Fibrovascular membrane                 | <0.001              | 0.22                  | 0.061–0.817| 0.02    |
| Retinal detachment                     | 0.001               | 1.23                  | 0.00       | 0.99    |
| Macular detachment                     | <0.001              | 0.25                  | 0.04–1.46  | 0.12    |
| Intraoperative retinal photocoagulation| 0.001               | 0.52                  | 0.11–2.36  | 0.39    |
| Gas tamponade                          | <0.001              | 0.39                  | 0.07–2.1   | 0.28    |
| Silicone oil tamponade                 | 0.04                | 0.00                  | 0.00       | 0.99    |
| Intraoperative complications           | 0.01                | 0.18                  | 0.03–1.02  | 0.06    |
| Reoperation                            | <0.001              | 0.21                  | 0.04–1.17  | 0.07    |

OR: Odds ratio; CI: Confidence interval.

**Table 5. Analysis of the factors correlated with the visual acuity of ≥20/30 at 2 years after surgery.**

| | Univariate analysis | Multivariate analysis | | |
|---|---|---|---|---|
| | P value | OR | 95% CI | P value |
| PT-INR value | 0.029 | 0.08 | 0.00–233.09 | 0.54 |
| APTT | 0.003 | 0.89 | 0.76–1.04 | 0.14 |
| Rubeosis iridis | 0.026 | 0.07 | 0.01–0.40 | 0.03 |
| Vitreous hemorrhage | < 0.001 | 9.55 | 1.03–95.27 | 0.04 |
| Fibrovascular membrane | 0.001 | 0.52 | 0.18–1.51 | 0.23 |
| Retinal detachment | 0.003 | 1.01 | 0.18–5.51 | 0.99 |
| Macular detachment | 0.002 | 0.25 | 0.04–1.46 | 0.12 |
| Intraoperative retinal photocoagulation | 0.001 | 0.58 | 0.22–1.48 | 0.25 |
| Gas tamponade | 0.002 | 0.29 | 0.06–1.45 | 0.13 |
| Reoperation | 0.001 | 0.08 | 0.05–1.18 | 0.07 |
| Postoperative complications | 0.04 | 0.45 | 0.11–1.89 | 0.28 |
| PT-INR value | 0.029 | 0.08 | 0.00–233.09 | 0.54 |
| APTT | 0.003 | 0.89 | 0.76–1.04 | 0.14 |

OR: Odds ratio; CI: Confidence interval.

Moreover, VH may occur from the neovascularization in the eye. Retinal function is not necessarily impaired by VH itself and is often maintained if no retinal detachment occurs. However, considering that the eye can hardly be observed because of VH occurrence, panretinal laser photocoagulation could not be performed [13]. VH is one of the factors to consider when making early decision to undergo vitreous surgery for PDR [15]. In the current study, the presence of VH is associated with the visual acuity of ≥20/30 at 2 years after the primary vitrectomy for PDR. This result may suggest that surgery without hesitation will provide a better therapeutic effect.

Meanwhile, fibrovascular membrane is formed with the neovascularization in contact with the retina through the epicenters. When the retina is pulled to the tangential direction by the fibrovascular membrane, tractional retinal detachment and macular detachment occur. When the traction in a thinned ischemic retina is excessive, retinal break may form, resulting in a combined tractional and rhegmatogenous retinal detachment. Hence, vitreous surgery should be performed as soon as possible [13]. La Heij et al. [16] observed 44 cases of 33 patients with tractional retinal detachment and macular detachment of PDR for median follow-up of 10 months retrospectively; they found that 22 eyes (50%) achieved a visual acuity of >20/200. In the present study, fibrovascular membrane correlated with the visual acuity of ≥20/40 after 2

**Table 6. Analysis of the factors correlated with the visual acuity of ≥20/40 at 4 years after surgery.**

| | Univariate analysis | Multivariate analysis | | |
|---|---|---|---|---|
| | | OR | 95% CI | P value |
| Ocular hypertension | 0.02 | 0.15 | 0.01–2.31 | 0.17 |
| Fibrovascular membrane | 0.003 | 0.26 | 0.07–0.94 | 0.04 |
| Retinal detachment | 0.002 | 0.85 | 0.18–4.07 | 0.84 |
| Macular detachment | 0.005 | 0.11 | 0.01–1.20 | 0.07 |
| Gas tamponade | 0.006 | 0.84 | 0.17–4.22 | 0.84 |
| Reoperation | 0.005 | 0.15 | 0.03–0.78 | 0.02 |

OR: Odds ratio; CI: Confidence interval.

**Table 7. Analysis of the factors correlated with the visual acuity of ≥20/30 at 4 years after surgery.**

| | Univariate analysis | Multivariate analysis | | |
| --- | --- | --- | --- | --- |
| | | OR | 95% CI | P value |
| Rubeosis iridis | 0.001 | 0.078 | 0.006–0.96 | 0.04 |
| Ocular hypertension | 0.004 | 1.01 | 0.05–20.59 | 0.99 |
| Vitreous hemorrhage | 0.001 | 0.66 | 0.1–4.41 | 0.67 |
| Fibrovascular membrane | <0.001 | 0.14 | 0.04–0.52 | 0.004 |
| Retinal detachment | 0.001 | 0.61 | 0.12–3.22 | 0.57 |
| Macular detachment | <0.001 | 0.18 | 0.01–3.41 | 0.25 |
| Intraoperative retinal photocoagulation | 0.001 | 0.24 | 0.05–1.28 | 0.96 |
| Gas tamponade | <0.001 | 0.67 | 0.09–4.59 | 0.69 |
| Reoperation | <0.001 | 0.06 | 0.07–0.54 | 0.01 |

OR: Odds ratio; CI: Confidence interval.

years, and ≥20/40 and ≥20/30 after 4 years postoperatively. Conversely, retinal detachment and macular detachment did not correlate with postoperative visual acuity, as confirmed by multivariate analysis. Considering that fibrovascular membrane is the cause of tractional retinal detachment and macular detachment, this condition might correlate with postoperative visual acuity stronger than retinal detachment and macular detachment.

Furthermore, the absence of reoperation only correlated with the visual acuity at 4 years postoperatively in this study. Vitreous surgery for PDR can lead to various complications in which the most common were corneal epithelial defects, elevated intraocular pressure, cataract formation, recurrent VH, rhegmatogenous retinal detachment, and neovascular glaucoma [17]. In this study, those with recurrent VH, retinal detachment, or neovascular glaucoma after primary vitrectomy for PDR underwent reoperation, indicating that performing minimal activity after surgery for PDR is required to maintain visual acuity from 2 years to 4 years.

The relationship between systemic factors and diabetic retinopathy has been reported extensively. Gupta et al. [9] showed that the factors related to poor visual outcome were longer duration of DM, use of insulin, delay in surgery, presence of ischemic heart disease, and failure to attend clinical appointments. According to Yorston et al. [11], systemic factors were not related to visual outcome in cases with vitrectomy for PDR. However, our study found an association between systemic factors and visual acuity at 2 and 4 years after a primary vitreous surgery for PDR. Therefore, the progress of vitreous surgery has improved the treatment results of PDR without being greatly affected by systemic factors.

Furthermore, visual acuity results after vitreous surgery for PDR had been previously reported. G Ratnarajan et al. [10] observed the VH cases of PDR in which 88 eyes of 80 patients (type 2 DM: 69 eyes of 62 cases, type 1 DM: 19 eyes of 18 cases) underwent 20-G vitreous surgery 44 months after the primary surgery. They found that the postoperative mean logMAR visual acuity was significantly improved (type 2 DM: logMAR from 0.64 to 0.46, type 1 DM: logMAR from 0.37 to 0.47). In addition, B Gupta et al. [9] observed the 346 eyes of 249 patients with traction retinal detachment and VH of PDR; the patients underwent 20-G vitreous surgery and were followed up for a mean of 1.44 years, and 39.4% of them achieved a final visual acuity of ≥20/40. D Yorston et al. [11] observed the 174 eyes of 148 patients who underwent 20-G vitreous surgery and were followed up for 8 months; an improvement in logMAR visual acuity of >0.3 was found in 74.4%. In our study, we included not only 20-G cases but also MIVS cases. The percentage of cases with a visual acuity of ≥20/40 was 64.0% (82 eyes), and that of cases with ≥20/30 visual acuity was 39.8% (51 eyes). This study also showed that an

improvement in logMAR visual acuity of >0.3 was found in 74.2% and 69.2% of patients at 2 and 4 years after a primary surgery, respectively. This result is similar to the result obtained from a previous study [11].

Although the number of cases in our study is smaller than that in previous reports, the original objectives of our study are to examine cases that could be completely followed up 2 or 4 years after a primary surgery and to reduce the number of dropout cases by collecting information on patients who have visited other ophthalmology clinics. No bias occurred because patient backgrounds had no significant difference between cases that could be followed up for 4 years and cases that could be followed up for 2–3 years after a primary surgery (Tables 1 and 2).

However, the current study has several limitations. This study has a retrospective design, and the surgical procedures were performed by two surgeons. Verification at another facility and prospective study are necessary to prove the usefulness of the factors clarified in this research. The surgical techniques of the two vitreoretinal surgeons might have minor differences, which may have introduced bias into the anatomic results and visual outcomes. Differences in therapeutic effects between 20, 23, and 25 gauges, which could not be examined in this study and in treatment outcomes due to proliferative membrane grading should be further evaluated.

In summary, the factors correlated with better postoperative visual acuity in PDR cases were as follows: no rubeosis iridis, no fibrovascular membrane, presence of VH, and no reoperation. Provision of treatment before the condition became severe and the lack of treatment complications were associated with good postoperative visual acuity. Furthermore, patients with PDR need to be treated at an early stage without missing the timing of surgical treatment.

## Supporting information

**S1 File. Supplementary excel file with raw data.**
(XLSX)

## Author Contributions

**Conceptualization:** Katsuhiro Nishi, Koichi Nishitsuka, Teiko Yamamoto, Hidetoshi Yamashita.

**Data curation:** Katsuhiro Nishi, Koichi Nishitsuka, Teiko Yamamoto.

**Formal analysis:** Katsuhiro Nishi, Koichi Nishitsuka.

**Funding acquisition:** Katsuhiro Nishi, Koichi Nishitsuka, Hidetoshi Yamashita.

**Investigation:** Katsuhiro Nishi, Koichi Nishitsuka.

**Methodology:** Katsuhiro Nishi, Koichi Nishitsuka.

**Project administration:** Katsuhiro Nishi, Koichi Nishitsuka.

**Resources:** Katsuhiro Nishi, Koichi Nishitsuka.

**Software:** Katsuhiro Nishi, Koichi Nishitsuka.

**Supervision:** Katsuhiro Nishi, Koichi Nishitsuka, Teiko Yamamoto, Hidetoshi Yamashita.

**Validation:** Katsuhiro Nishi, Koichi Nishitsuka.

**Visualization:** Katsuhiro Nishi, Koichi Nishitsuka.

**Writing – original draft:** Katsuhiro Nishi, Koichi Nishitsuka, Hidetoshi Yamashita.

**Writing – review & editing:** Katsuhiro Nishi, Koichi Nishitsuka, Hidetoshi Yamashita.

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
