## [Decision Letter · Decision Letter 0]

5 Oct 2020

PONE-D-20-28142

Factors correlated with visual outcomes at two and four years after vitreous surgery for proliferative diabetic retinopathy

PLOS ONE

Dear Dr. Nishitsuka,

Thank you for submitting your manuscript to PLOS ONE. After careful consideration, we feel that it has merit but does not fully meet PLOS ONE’s publication criteria as it currently stands. Therefore, we invite you to submit a revised version of the manuscript that addresses the points raised during the review process.

We look forward to receiving your revised manuscript.

Kind regards,

Andrzej Grzybowski

Academic Editor

PLOS ONE

Journal Requirements:

Reviewers' comments:

Reviewer's Responses to Questions

**Comments to the Author**

1. Is the manuscript technically sound, and do the data support the conclusions?

Reviewer #1: Yes

Reviewer #2: Partly

2. Has the statistical analysis been performed appropriately and rigorously? 

Reviewer #1: Yes

Reviewer #2: Yes

3. Have the authors made all data underlying the findings in their manuscript fully available?

Reviewer #1: Yes

Reviewer #2: No

4. Is the manuscript presented in an intelligible fashion and written in standard English?

Reviewer #1: Yes

Reviewer #2: Yes

5. Review Comments to the Author

Reviewer #1: THis is a nice set of data regarding long-term outcomes for patients undergoing vitrectomy surgery for proliferative diabetic retinopathy. A few clarifications would be helpful:

1) Can the authors please describe the lens status of the patients? Did all patients receive cataract surgery at the time of vitrectomy? Were some already pseudophakic? Did any receive cataract surgery at a later time after the vitrectomy? (I ask, because cataract progression could affect final outcomes).

2) Did any patients receive anti-VEGF as a pre-operative adjunct to the surgery to reduce bleeding?

3) Did any patients develop macular edema after the vitrectomy requiring anti-VEGF therapy?

4) Of the patient with pre-operative rubeosis, did any need to go on to glaucoma surgery? What was the pre-vitrectomy stage of the rebeosis (early versus late stage)?

5) Can the authors provide some level of quantification of the proliferative membranes, and whether the extent of proliferation correlated with outcomes?

Reviewer #2: Dear authors

I would like to congratulate you on your work. The findings and results are important and can help surgeons make the right decisions. The advantages of the study are: large cohort of patients, long follow up and the fact that only two vitreoretinal surgeons performed surgeries.

The study has some major limitations:

• Effect of gauges on vitreoretinal surgery: There is considerable talk about functional and morphological results after vitreoretinal surgery performed with the help of 20G, 23 G , 25 G but no analysis on the effect of the different gauges. Did the evolution of vitreoretinal surgery have any effect on the results?

• No information on treatment regiments before and after surgery: Patients with PDR will most certainly need treatment with injection and perhaps laser treatment before and after surgery. There is only one short mention on the use of Triamncinolone before surgery but no mention on the use of injections or laser before and after surgery. I believe this is a very important part that will judge the anatomical and functional results and has to be mentioned with sub-analysis in each group. (lower initial BCVA means more or less post OP treatment with injections or laser?)

• Did the status of the fibrovascular membranes and the status of the posterior hyaloid play a role in the functional and morphological results? Did the manipulation of these mambranes have an effect on the RD rate and final BCVA?

• Lines 111 and 112 Explain: Among them, 91 eyes of 70 patients could be followed up for 4 years after 112 a primary surgery, and 37 eyes of 30 patients could be followed up for 2-3 years How is it possible to have data on 91 eyes with a 4 year follow up and only 37 eyes with a follow up up to 2-3 years. Is it opposite correct? What do you mean with 2-3 years? Not precise enough

• Lines 153 to 160: A statistical association between initial BCVA and improvement of vision after 2-3 year and 4 year is essential! Which patients profited more from surgery? Does initial BCVA is a prognostical factor?

• Discussion Line 214-220: Important and helpful results but these results are incomplete if the treatment regiments before and after surgeries are not elaborated (did patients with rubeosis iridis receive any or more post OP injections or laser treatments compared to patients with no rubeosis?)

6. PLOS authors have the option to publish the peer review history of their article (what does this mean?). If published, this will include your full peer review and any attached files.

Reviewer #1: No

Reviewer #2: No

---

## [Author Response · Author response to Decision Letter 0]

14 Nov 2020

Reviewer #1: THis is a nice set of data regarding long-term outcomes for patients undergoing vitrectomy surgery for proliferative diabetic retinopathy. 

Response: Thank you for cautiously reviewing our manuscript. We found that there was a minor error in Figure 1. Hence, it was modified. Moreover, some data in the Result section were revised. Nevertheless, the changes did not affect the multivariate analysis in this study.

Two years after surgery, the visual acuity values in 64.0% (82 of 128 eyes) and 39.8% (50 eyes of 128 eyes) of patients were ≥20/40 and ≥20/30, respectively. Four years after surgery, the visual acuity values in 53.8% (49 of 91 eyes) and 46.1% (42 of 91 eyes) of patients were ≥20/40 and ≥20/30, respectively (page 13, lines 149–152).

A few clarifications would be helpful:

1) Can the authors please describe the lens status of the patients? Did all patients receive cataract surgery at the time of vitrectomy? Were some already pseudophakic? Did any receive cataract surgery at a later time after the vitrectomy? (I ask, because cataract progression could affect final outcomes).

Response: Thank you for the valuable comment. The lens status of patients is presented in Table 1. In total, 63 of 128 eyes underwent cataract surgery during vitrectomy. Moreover, 38 of 128 eyes were pseudophakic during treatment. This study mainly assessed factors correlated to long-term visual acuity prognosis and the initial treatment of PDR. During the course, the attending physician treated macular edema, cataract, and late cataract if necessary. The following data were added in the Material and Methods section:

Participants were treated for vision-affecting lesions such as posterior capsular opacification, progressed cataract, neovascular glaucoma, and diabetic macular edema (DME) during the postoperative course (page 5, lines 73–75).

2) Did any patients receive anti-VEGF as a pre-operative adjunct to the surgery to reduce bleeding?

Response: Thank you for this valuable comment. As shown in page 5, line 70, all patients did not receive anti-vascular endothelial growth factor (VEGF) therapy during vitrectomy. This information was modified as follows:

All patients did not receive anti-vascular endothelial growth factor (VEGF) therapy as a preoperative adjunct (page 5, lines 70–72).

3) Did any patients develop macular edema after the vitrectomy requiring anti-VEGF therapy?

Response: Thank you for this valuable comment. In total, 11 patients presented with DME that required additional treatment after vitrectomy. During the research period, VEGF therapy was not approved in Japan and patients were treated with retinal photocoagulation and local steroid. The following data were added in the Result section: 

In total, 11 patients presented with diabetic macular edema that required additional treatment after vitrectomy. During the research period, VEGF treatment was not approved in Japan, and patients were treated with retinal photocoagulation and local steroid (page 8, lines 124–127).

.

4) Of the patient with pre-operative rubeosis, did any need to go on to glaucoma surgery? What was the pre-vitrectomy stage of the rebeosis (early versus late stage)?

Response: Thank you for this valuable comment. This study investigated the relationship between preoperative rubeosis and long-term visual acuity prognosis. PRP was cautiously performed during vitrectomy for rubeosis in PDR. Postoperatively, NVG was found in seven eyes and glaucoma surgery was performed. Postoperative NVG eye counts and treatment policies are described as follows:

Pan retinal photocoagulation was cautiously performed before or during vitrectomy on all patients (page 5, lines 72–73).

Seven eyes presented with neovascular glaucoma that required additional treatment after vitrectomy (page 8, lines 127–128).

5) Can the authors provide some level of quantification of the proliferative membranes, and whether the extent of proliferation correlated with outcomes?

Response: Thank you for this valuable comment. The relationship between proliferative membrane grading and surgical outcomes is also interesting. This association could not be assessed using the current data. However, our protocols could be modified for a follow-up study. This information has been acknowledged and it can be used as a topic in further research. These data were included in the Discussion section of the revised manuscript.

Differences in therapeutic effects between 20, 23, and 25 gauges, which could not be examined in this study and in treatment outcomes due to proliferative membrane grading should be further evaluated (page 24, lines 307–309).

Reviewer #2: Dear authors

I would like to congratulate you on your work. The findings and results are important and can help surgeons make the right decisions. The advantages of the study are: large cohort of patients, long follow up and the fact that only two vitreoretinal surgeons performed surgeries.

Response: Thank you for cautiously reviewing our manuscript. We found that there was a minor error in Figure 1. Hence, it was revised. Moreover, some data in the Result section were modified. However, these changes did not affect the multivariate analysis in this study. 

Two years after surgery, the visual acuity values in 64.0% (82 of 128 eyes) and 39.8% (50 of 128 eyes) of patients were ≥20/40 and ≥20/30, respectively. Four years after surgery, the visual acuity values in 53.8% (49 of 91 eyes) and 46.1% (42 of 91 eyes) of patients were ≥20/40 and ≥20/30, respectively (page 13, lines 149–152).

.

The study has some major limitations:

• Effect of gauges on vitreoretinal surgery: There is considerable talk about functional and morphological results after vitreoretinal surgery performed with the help of 20G, 23 G , 25 G but no analysis on the effect of the different gauges. Did the evolution of vitreoretinal surgery have any effect on the results?

Response: Thank you for this valuable comment. The effect of vitreoretinal surgery (MIVS) evolution is also interesting. In this study, the number of cases in each gauge was not sufficient. Hence, we are considering 57 eyes in MIVS (23G + 25G) and 71 eyes in 20 gauge. Tables 4–7 show the factors correlated to postoperative visual acuity based on the univariate and multivariate analyses. However, MIVS did not affect visual acuity. In future studies, the number of cases should be increased, and each gauge must be analyzed. This information has been acknowledged and it can be used as a topic in further research. These data were included in the Discussion section of the revised manuscript.

Differences in therapeutic effects between 20, 23, and 25 gauges, which could not be examined in this study and in treatment outcomes due to proliferative membrane grading should be further evaluated. (page 24, lines 307–309).

• No information on treatment regiments before and after surgery: Patients with PDR will most certainly need treatment with injection and perhaps laser treatment before and after surgery. There is only one short mention on the use of Triamncinolone before surgery but no mention on the use of injections or laser before and after surgery. I believe this is a very important part that will judge the anatomical and functional results and has to be mentioned with sub-analysis in each group. (lower initial BCVA means more or less post OP treatment with injections or laser?)

Response: Thank you for the valuable comment. The protocol for PDR surgery in this study is depicted in the Material and Methods section. 

All patients did not receive anti-vascular endothelial growth factor (VEGF) therapy as a preoperative adjunct. Pan retinal photocoagulation was cautiously performed before or during vitrectomy on all patients. Participants were treated for vision-affecting lesions such as posterior capsular opacification, progressed cataract, neovascular glaucoma, and diabetic macular edema (DME) during the postoperative course (page 5, lines 70–75).

One patient who received preoperative injection of triamcinolone had a history of using triamcinolone for diabetic macular edema before PDR treatment. This information was revised as follows:

Moreover, the ophthalmologic findings were categorized into three sections: preoperative, intraoperative, and postoperative. The preoperative ophthalmologic findings were as follows: intraocular lens implantation, retinal photocoagulation, the history of intravitreal injection of triamcinolone acetonide, rubeosis iridis, ocular hypertension (>21 mmHg), VH, posterior vitreous detachment, fibrovascular membrane, retinal detachment, and macular detachment (page 6, lines 83–88).

.

The low preoperative visual acuity may be attributed to PDR or vitreous hemorrhage. Vitreous hemorrhage affecting long-term visual acuity may be attributed to mild conditions, and the surgery was timely performed. These data were added in the Discussion section (page 21, lines 242–249).

• Did the status of the fibrovascular membranes and the status of the posterior hyaloid play a role in the functional and morphological results? Did the manipulation of these mambranes have an effect on the RD rate and final BCVA?

Response: Thank you for the valuable comment. Tables 4–7 show the factors correlated to postoperative visual acuity based on the univariate and multivariate analyses. However, posterior vitreous detachment did not affect visual acuity. The relationship between proliferative membrane grading and surgical outcomes is also interesting. This association could not be assessed using the current data. However, protocols could be modified for a follow-up study. This information has been acknowledged, and it can be used as a topic in further research. These data were added in the Discussion section of the revised manuscript.

Differences in therapeutic effects between 20, 23, and 25 gauges, which could not be examined in this study and in treatment outcomes due to proliferative membrane grading should be further evaluated (page 24, lines 307–309).

• Lines 111 and 112 Explain: Among them, 91 eyes of 70 patients could be followed up for 4 years after 112 a primary surgery, and 37 eyes of 30 patients could be followed up for 2-3 years How is it possible to have data on 91 eyes with a 4 year follow up and only 37 eyes with a follow up up to 2-3 years. Is it opposite correct? What do you mean with 2-3 years? Not precise enough

Response: Thank you for the valuable comment. As depicted in the first paragraph of the Result section, 128 eyes from 100 patients could be followed-up for >2 years after the primary surgery. That is, 91 eyes from 70 patients could be evaluated 4 years after the primary surgery. However, 37 eyes from 30 patients could not be followed-up 4 years after the primary surgery. Therefore, 37 eyes were categorized under the 2–3-year follow-up group. This information was revised as follows:

In total, 91 eyes from 70 patients could be followed-up 4 years after the primary surgery. However, 37 eyes from 30 patients could not be evaluated 4 years after the primary surgery. Therefore, 37 eyes were categorized under the 2–3-year follow-up group (page 8, lines 116–118).

• Lines 153 to 160: A statistical association between initial BCVA and improvement of vision after 2-3 year and 4 year is essential! Which patients profited more from surgery? Does initial BCVA is a prognostical factor?

Response: Thank you for this valuable comment. The low preoperative visual acuity might be attributed to PDR or vitreous hemorrhage. Vitreous hemorrhage affecting long-term visual acuity might be caused by mild conditions, and surgery was timely performed. These data were included in the Discussion section. The current study showed that a low preoperative visual acuity is not a serious condition.

• Discussion Line 214-220: Important and helpful results but these results are incomplete if the treatment regiments before and after surgeries are not elaborated (did patients with rubeosis iridis receive any or more post OP injections or laser treatments compared to patients with no rubeosis?)

Response: Thank you for this valuable comment. During the research period, VEGF therapy was not approved in Japan, and patients were treated with retinal photocoagulation and local steroid treatment. The protocol for PDR surgery in this study is depicted in the Method and Materials section. 

All patients did not receive anti-vascular endothelial growth factor (VEGF) therapy as a preoperative adjunct. Pan retinal photocoagulation was cautiously performed before or during vitrectomy on all patients. Participants were treated for vision-affecting lesions such as posterior capsular opacification, progressed cataract, neovascular glaucoma, and diabetic macular edema (DME) during the postoperative course (page 5, lines 70–75).

---

## [Decision Letter · Decision Letter 1]

8 Dec 2020

Factors correlated with visual outcomes at two and four years after vitreous surgery for proliferative diabetic retinopathy

PONE-D-20-28142R1

Dear Dr. Nishitsuka,

We’re pleased to inform you that your manuscript has been judged scientifically suitable for publication and will be formally accepted for publication once it meets all outstanding technical requirements.

Kind regards,

Andrzej Grzybowski

Academic Editor

PLOS ONE

Additional Editor Comments (optional):

I agree with Reviewer 3, who confirms the original concerns noted by the reviewers have been adequately addressed.

Reviewers' comments:

Reviewer's Responses to Questions

**Comments to the Author**

1. If the authors have adequately addressed your comments raised in a previous round of review and you feel that this manuscript is now acceptable for publication, you may indicate that here to bypass the “Comments to the Author” section, enter your conflict of interest statement in the “Confidential to Editor” section, and submit your "Accept" recommendation.

Reviewer #1: All comments have been addressed

Reviewer #2: (No Response)

Reviewer #3: All comments have been addressed

2. Is the manuscript technically sound, and do the data support the conclusions?

Reviewer #1: Yes

Reviewer #2: Partly

Reviewer #3: Yes

3. Has the statistical analysis been performed appropriately and rigorously? 

Reviewer #1: Yes

Reviewer #2: I Don't Know

Reviewer #3: Yes

4. Have the authors made all data underlying the findings in their manuscript fully available?

Reviewer #1: Yes

Reviewer #2: Yes

Reviewer #3: Yes

5. Is the manuscript presented in an intelligible fashion and written in standard English?

Reviewer #1: Yes

Reviewer #2: Yes

Reviewer #3: Yes

6. Review Comments to the Author

Reviewer #1: Thank you for addressing my comments. I appreciate the clarifications and feel that your work adds new information to the literature.

Reviewer #2: Dear authors.

Thank you for the revision of your work. One major limitation of the work is the absence of anti VEGF parameters. The injection of steroids is not up to date. The major advantage of the work is the long term follow up. I regret to inform you that i would not recommend this papaer to be published.

Reviewer #3: This revised manuscript explores factors associated with better vision outcomes after vitrectomy for proliferative diabetic retinopathy (PDR). As the authors point out, patients with PDR often have vitrectomy in their 50's, but go on to live to an average age of 70. Therefore it is important to determine factors associated with preserved vision after vitrectomy for PDR. A total of 128 eyes in 100 patients operated between January 2008 and September 2012, followed for 2 to 4 years or more were evaluated. Of these, 91 eyes from 70 patients were followed for 4 years. Those followed for more than 2 years, but less than 4 years were described as a group followed for 2-3 years. Factors associated with better vision included:

1. No preoperative rubeosis, which, if present, would be a sign of severe retinal ischemia.

2. No preoperative fibrovascular membrane.

3. Preoperative vitreous hemorrhage did not adversely affect vision.

4. No reoperation.

Questions from two reviewers of the original manuscript were answered satisfactorily. The study was carried out in Japan, where anti-VEGF therapy was not approved, and was therefore unavailable during the study. Panretinal photocoagulation was utilized for retinal neovascularization and rubeosis. Triamcinolone and focal macular laser were used for diabetic macular edema.

7. PLOS authors have the option to publish the peer review history of their article (what does this mean?). If published, this will include your full peer review and any attached files.

Reviewer #1: No

Reviewer #2: No

Reviewer #3: No

---

## [Editor Report · Acceptance letter]

6 Jan 2021

PONE-D-20-28142R1 

Factors correlated with visual outcomes at two and four years after vitreous surgery for proliferative diabetic retinopathy 

Dear Dr. Nishitsuka:

I'm pleased to inform you that your manuscript has been deemed suitable for publication in PLOS ONE. Congratulations! Your manuscript is now with our production department. 

Kind regards, 

on behalf of

Dr. Andrzej Grzybowski 

Academic Editor

PLOS ONE